# Molecular Characterization of the Response to Conventional Chemotherapeutics in Pro-B-ALL Cell Lines in Terms of Tumor Relapse

**DOI:** 10.3390/genes13071240

**Published:** 2022-07-14

**Authors:** Yvonne Saara Gladbach, Lisa-Madeleine Sklarz, Catrin Roolf, Julia Beck, Ekkehard Schütz, Georg Fuellen, Christian Junghanss, Hugo Murua Escobar, Mohamed Hamed

**Affiliations:** 1Institute for Biostatistics and Informatics in Medicine and Ageing Research (IBIMA), Rostock University Medical Center, 18057 Rostock, Germany; y.s.gladbach@hotmail.de (Y.S.G.); fuellen@uni-rostock.de (G.F.); 2Faculty of Biosciences, Heidelberg University, 69120 Heidelberg, Germany; 3Division of Applied Bioinformatics, German Cancer Research Center (DKFZ), 69120 Heidelberg, Germany; 4Clinic III—Hematology, Oncology, Palliative Medicine, Center for Internal Medicine, Rostock University Medical Center, 18057 Rostock, Germany; lisa-madeleine.sklarz@uni-rostock.de (L.-M.S.); catrin.roolf@gmail.com (C.R.); christian.junghanss@med.uni-rostock.de (C.J.); hugo.murua.escobar@med.uni-rostock.de (H.M.E.); 5Chronix Biomedical GmbH, 37073 Göttingen, Germany; jbeck@chronixbiomedical.de (J.B.); esc@chronixbiomedical.de (E.S.); 6Comprehensive Cancer Center Mecklenburg-Vorpommern (CCC-MV), Campus Rostock, Rostock University Medical Center, 18057 Rostock, Germany

**Keywords:** drug response, tumor relapse, acute lymphoblastic leukemia, cytostatics, cytarabine, dexamethasone

## Abstract

Little is known about optimally applying chemotherapeutic agents in a specific temporal sequence to rapidly reduce the tumor load and to improve therapeutic efficacy. The clinical optimization of drug efficacy while reducing side effects is still restricted due to an incomplete understanding of the mode of action and related tumor relapse mechanisms on the molecular level. The molecular characterization of transcriptomic drug signatures can help to identify the affected pathways, downstream regulated genes and regulatory interactions related to tumor relapse in response to drug application. We tried to outline the dynamic regulatory reprogramming leading to tumor relapse in relapsed MLL-rearranged pro-B-cell acute lymphoblastic leukemia (B-ALL) cells in response to two first-line treatments: dexamethasone (Dexa) and cytarabine (AraC). We performed an integrative molecular analysis of whole transcriptome profiles of each treatment, specifically considering public knowledge of miRNA regulation via a network-based approach to unravel key driver genes and miRNAs that may control the relapse mechanisms accompanying each treatment. Our results gave hints to the crucial regulatory roles of genes leading to Dexa-resistance and related miRNAs linked to chemosensitivity. These genes and miRNAs should be further investigated in preclinical models to obtain more hints about relapse processes.

## 1. Introduction

Acute lymphoblastic leukemia (ALL) is a heterogeneous disease characterized by a clonal proliferation of lymphoid precursor cells of the bone marrow, most commonly B-cells (B-ALL) [1,2,3,4]. Children with ALL have a long-term survival of 80% [4,5], whereas adults show a <30% disease-free survival [6]. Besides age, genetic rearrangements such as translocations indicate higher chances for relapse and a poor overall survival rate. Patients harboring a rearrangement of the histone-lysine N-methyltransferase 2A (KMT2A) gene on chromosome 11 have a poorer prognosis compared to patients without [7,8,9] due to resistance to treatment [10].

Different chemotherapeutic agents were used to reduce the amount of leukemic cells and to amend clinical protocols of radiation, immunotherapy and transplantation. The clinical protocols depend on factors such as the vital state of the patient, risk stratification of the different leukemic subtypes, and response to therapy. Several agents are currently applied using different protocols and show remarkable effectivity in the clinics. Nevertheless, it is possible that patients are affected by remaining malignant cells caused by treatment resistance leading finally to a relapse [7,11,12,13].

The treatment regimens of adult ALL were adapted from childhood ALL [14] and chronic lymphoblastic leukemia (CLL) protocols [11]. These therapeutic protocols suggested the application of several cytostatic agents, including glucocorticoids (GCs) such as dexamethasone (Dexa), as well as cytarabine (AraC) [1,15,16]. The GC Dexa is a long-established agent in the current treatment regimen [17,18,19] and binds to specific cytoplasmic GC receptors. It mediates the inflammatory response by inhibiting leukocyte infiltration at the inflammation site [19], acts as an anti-inflammatory drug, is used in cancer therapies and has a pro-apoptotic effect on malignancies with lymphoproliferative disorders. AraC is a nucleoside analog and it is incorporated into the DNA to directly inhibit the DNA polymerase, affecting DNA synthesis in the S phase of the cell cycle [20]. This makes AraC a main agent for the induction and extra-compartment therapy [21], either as a monotherapy or in combination with targeted agents [16]. While Dexa and AraC are highly effective in B-ALL treatment, their efficacy is still limited by severe adverse effects and a high relapse incidence [7]. For instance, 20% of B-ALL patients who are treated by GCs relapse and die from the disease [20], and survivors often suffer lifelong adverse effects [22,23]. Often, therapeutic success is limited, caused by remaining minimal residual disease after transplantation and/or chemotherapy [13] and the development of drug resistance [7]. Therefore, understanding relapse mechanisms is important for the choice of the clinical protocols and for developing superior treatment protocols.

The tumor escape mechanisms are not well-characterized yet in B-ALL [24]. Useful insights can be gained by understanding the mode of action of therapeutic agents on pro-B-ALL cell lines such as SEM and RS4;11. RS4;11 is a good model to investigate adult relapsed forms of KMT2A-rearranged pro-B-ALL, while SEM is suitable for the study of the childhood type [25]. Apart from their KMT2A-rearrangement, both cell lines were chosen specifically because of their isolation after relapse and their respective resistance to AraC (RS4;11) or Dexa (SEM). 

Therefore, both cell lines were previously used to elucidate relapse-specific gene expression signatures [26]. These cells were previously incubated with AraC, as well as other nucleotide analogs, such as decitabine, zebularine, gemcitabine, and floxuridine [27,28]. Other studies incubated these cells with the GCs Dexa and prednisolone [29] to study the molecular mechanisms of GC-resistance [30,31]. 

To better understand the regulatory reprogramming associated with tumor relapse in response to Dexa and AraC, we here report a detailed investigation of the dynamic molecular responses of RS4;11 and SEM to both treatments. Cell-biological assays analyzing cell proliferation and metabolism revealed resistance mechanisms for AraC in RS4;11 and Dexa in SEM, which was as expected. We jointly analyzed the whole transcriptome profiles describing the effects of the two treatments and potential miRNA regulators, and we employed publicly available regulatory databases in order to gain a deeper understanding of regulatory mechanisms in driving the relapse process associated with the respective drug treatment. The cooperative functional role of the co-regulated genes during the relapse process was evaluated in terms of statistical significance and biological relevance. We also characterized the exclusive gene signatures regulated by the corresponding drug and the associated functional terms and pathways enriched in each cell line. 

## 2. Materials and Methods

### 2.1. Cell Lines and Culture Conditions

Both pro-B-ALL cell lines were purchased from “Deutsche Sammlung von Mikroorganismen und Zellkulturen GmbH” (Braunschweig, Germany) and cultured according to the manufacturer’s protocol. RS4;11 was grown in Alpha-MEM Medium (Biochrom GmbH, Berlin, Germany) and SEM in Iscove’s MEM Medium (Biochrom GmbH, Berlin, Germany). Both media were supplemented with 1% penicillin/streptomycin (Biochrom GmbH, Berlin, Germany) and 10% heat-inactivated fetal bovine serum (Biochrom GmbH, Berlin, Germany). Cells were cultivated in a humidified atmosphere with 5% CO_2_ at 37 °C and placed in T175 tissue culture flasks (Greiner Bio-One GmbH, Frickenhausen, Germany) in downright position.

### 2.2. Cytostatic Agents

Cytarabine (AraC) (100 mg/mL) was purchased from cell pharm GmbH (Bad Vilbel, Germany) and Dexamethasone (Dexa) (8 mg inject Jenapharm^®^) from mibe GmbH (Brehna, Germany). The cytostatics were diluted in phosphate-buffered saline (PBS) for the inhibitory experiments. Control cells were incubated with their respective medium containing the same concentration of PBS as the cells treated with the different drugs. 

### 2.3. Drug Application Experiments

The cell lines were treated with drug concentrations similar to doses used in clinical settings. These concentrations allow for a threshold of above 30% living cells after 72 h of incubation or, alternatively, with a maximum of 10 µM. RS4;11 was incubated with either 10 µM AraC or 0.01 µM Dexa. The cell line SEM was incubated with either 0.1 µM AraC or 10 µM Dexa. For all experiments, a density of 3.33 × 10^5^ cells/mL was used. 

After 72 h incubation, cell biological analyses were performed to observe the effect on proliferation (trypan blue staining), proliferation and metabolism (WST-1 assay). Cell pellets were collected for further RNA-isolation and transcriptome sequencing. At this time point, another aliquot of the cells received new media and were incubated for another 72 h before the same readout was performed (readout 144 h). 

### 2.4. Biological Assays (Cell Count, Proliferation and Metabolic Activities) 

For the cell count analysis, RS4;11 and SEM cells were seeded at a density of 0.5 × 10^6^ cells per 1.5 mL in 24-well plates (Greiner Bio One GmbH, Frickenhausen, Germany) at the initial time point zero (for readout at 72 h) and, after 72 h (for readout at 144 h), cells were harvested and washed in PBS (10 min, 180 × g, 4 °C), and the cell counts were determined using trypan blue staining (Sigma-Aldrich Chemie GmbH, Steinheim, Germany). An overview of the experimental setup is presented in Figure 1.

For the WST-1 proliferation assay, both cell lines were seeded in biological triplicates at a density of 5 × 10^4^ cells per 150 µL per well in 96-well plates at the initial time point zero (readout 72 h) and after another 72 h (readout 144 h). The metabolic activity was analyzed via tetrazolium compound WST-1 (TaKaRa Bio Inc., Kusatsu, Japan), an indicator assay for the cells’ metabolic activity in comparison to the cell count. After 72 h and 144 h, the cells were incubated with 15 µL WST-1 for up to 3 h. The principle of this assay was based on the mitochondrial dehydrogenases of vital cells, which reduce the soluble WST-1 in the dark red formazan. The amount of formazan dye directly correlates to the activity and number of active metabolic cells and can be measured by photometer (absorbance: 450 nm; reference: 750 nm). As background control, we used the absorbance of pure culture medium with added WST-1. The comparison of the data of the WST-1 assay and the cell count gives hints about the metabolic activity of the cells. 

### 2.5. RNA-Isolation and Sequencing

Control cells were cultivated in T75- tissue culture flasks and Dexa or AraC incubated cells of both cell lines were cultivated in T175-tissue culture flasks in downright position. Cells were harvested and washed (10 min, 180 × g, 4 °C) in PBS three times. Total RNA was extracted using the miRNeasy Mini Kit (Qiagen, Hilden, Germany), as described in the Quick Start Protocol. Then, 500 µL Buffer RWT and 350 µL Buffer RPE were added twice after performing the DNase (Qiagen, Hilden, Germany) digest. The RNA quantity was assessed using the NanoDrop Spectrophotometer ND 1000 (Peqlab Biotechnologie GmbH, Erlangen, Germany, Version 3.7.1). For each condition/experiment, three biological replicates were prepared and a total of 1 µg total RNA, with RNA integrity numbers > 8 with poly-A enrichment, was used for preparing the sequencing libraries using the NEBNext Ultra RNA preparation Kit (New England Biolabs, Ipswich, MA, USA) according to the manufacturer’s protocols. The sequencing was conducted on an Illumina NextSeq500 (Illumina, San Diego, CA, USA) as single reads with 75 bp length. 

### 2.6. Data Processing 

FastQC v.0.11.5 [32] was used to evaluate the quality of the raw sequencing reads and adjust the correct adapter trimming. Trimmomatic v. 0.36 [33], with the following parameters: ILLUMINACLIP:./adapters/list.fa:2:30:10:6 LEADING:3 TRAILING:3 SLIDINGWINDOW:4:15 MINLEN:36, was applied for trimming the adapters and primers sequences. After the adapter trimming, sequence quality was checked with FastQC [32], and reads with an average Phred quality score of ≥30 were considered high quality. Next, all high-quality reads were mapped to the human reference genome GRCh38.78 (Ensembl) using TopHat v.2.1.1 [34]. With the combination of SAMtools v.1.3 [35] and HTSeq-count v.0.6.1p1 [36], the read counts of the mapped reads were counted per ENSEMBL-ID of the GRCh38.78. ENSEMBL-IDs were then mapped using the Bioconductor package org.Hs.eg.db v.3.5.0 [37] to the corresponding gene symbol. Differential expression gene (DEG) analysis was performed using the Bioconductor package DESeq2 v.1.12.4 [38] with an FDR threshold of 5% and log2 fold change of ± 1.5. 

### 2.7. Over-Representation Analysis (ORA)

An over-representation analysis was applied to the differentially expressed genes (DEGs) to filter out genes related to functional terms of stress release/response and metabolism (Appendix A) and to highlight the enriched GO terms and pathways in each drug response. ORA was performed using DAVID as previously described by Hamed et al., 2018 [39]. Statistical significance of the enriched terms and pathways was evaluated via Fisher’s exact test followed by the Benjamini–Hochberg adjustment [40] for controlling the false discovery rate (FDR), with a cutoff value of 0.05.

### 2.8. miRNA Enrichment Analysis and Related ORA

The miRNAs included in the GRNs (see Section 2.9) were predicted by identifying the set of miRNAs whose target genes are statistically enriched within the DEG lists (see Section 2.6). The miRNA target-genes associations were retrieved from the regulatory databases of TFmiR [41]. The hypergeometric test followed by Benjamini–Hochberg adjustment (FDR = 5%) was used to evaluate the significance of the miRNA enrichment. The over-representation analysis for each miRNA set was performed using TAM v2.0 [42,43] to identify the functional terms and diseases enriched in each miRNA set.

### 2.9. Gene Regulatory Network (GRN) Construction

The TFmiR web service [41] was used to construct the GRN for each drug treatment based on the DEG set and the corresponding enriched miRNA set. We considered only the regulatory interactions that are supported by experimental evidence. We contextualized the resulting networks to leukemia by considering only regulatory interactions whose source nodes or target nodes are known to be associated with leukemia, generating a leukemia-specific network for each drug treatment. Key network drivers (central hub miRNAs/genes) were identified using the union set of nodes considering each of four network centrality measures (degree, betweenness, closeness and eigenvector) [41,44]. More specifically, for each centrality measure, we selected the top 10% genes/miRNAs of highest centrality. The GRN networks were visualized using Cytoscape v3.3.0 [45]. 

### 2.10. Semantic Validation of the GRNs Nodes

The GoSemSim R package [46] was used to compute the functional similarity scores within the GRN nodes based on their GO annotation. Nominal *p*-values for evaluating the statistical significance of the functional homogeneity of the GRN genes were calculated based on 100 random gene sets. The Kolmogorov–Smirnov test was adopted to check whether the similarity scores of GRN gene pairs were statistically higher than the scores of randomly selected pairs. The FDR was controlled using the Benjamini–Hochberg procedure [40] with a cutoff value of 0.05.

## 3. Results

### 3.1. Experimental Design and Cell Line Characteristic Biological Effects of the Used Agents

To reflect patient-relevant settings, we exposed the cell lines RS4;11 and SEM to the drugs AraC and Dexa for 72 h, followed by 72 h without chemotherapeutic pressure (drug release, noted here as 144 h+/−); see Figure 1. This setting could help to identify potential driver genes, which are responsible for relapse in response to AraC or Dexa. 

In order to quantify the biological effects of the two drug applications on proliferation rate and metabolism, we set (1) the initially seeded cells as 100% in Figure 2a,d and (2) the control cells as 100% (readout 72 h and 144 h) in Figure 2b,c,e,f. In general, incubation with AraC and Dexa causes, in both cell lines, a reduced proliferation rate compared to the control cells after 72 h. The exposure of AraC in RS4;11 approximately halved the proliferation rate compared to the control. In contrast, the exposure of Dexa reduced the proliferation to 15% in comparison to the control. A low increase in metabolic activity was detected by both AraC and Dexa exposures. After 144 h in total, the AraC release (AraC +/− 144 h) resulted in nearly the same proliferation rate as the control cells, but Dexa resulted in an almost halved proliferation rate. The measurement of the metabolic activity showed, in both settings, comparable results to the proliferation rate (Figure 2c).

An AraC exposition in SEM leads to an approximately 15% proliferation rate, and Dexa to a < 10% proliferation rate compared to control cells. Comparable results were observed by measurements of the metabolic activity. The drug release resulted, after a total of 144 h, in the case of AraC, in approximately half of the proliferation rate of the control cells, but the Dexa release resulted in a quarter of the proliferation rate. Furthermore, the metabolic activity in AraC-exposed cells resulted in an increase compared to the results of the cell count (Figure 2f). 

### 3.2. Whole-Genome Transcriptomic Differences between AraC and Dexa 

To better elucidate the results of the biological assays on the molecular level, we performed a high coverage whole transcriptomic analysis of the RS4;11 and SEM cells following Dexa and AraC treatments. The PCA of the expression profiles for each drug treatment showed a clear separation between the drug responses of each cell line (Figure 3A). Interestingly, within each cell line, the samples after the 72 h drug application, as well as the corresponding samples after 72 h of drug release (144 h+/−), were clustered in different groups for each drug, indicating differences in the relapse-associated molecular response. The comparative RNA-Seq analysis of 72 h and 144 h+/− revealed genes that are dysregulated due to the stress release and metabolism, as well as potential genes involved in relapse in response to AraC and Dexa. We filtered out genes related to stress release and metabolism (see Methods) to focus our downstream analysis on relapse-associated genes. Considering the drug resistance and sensitivity of the two cell lines, marked differences in the genes potentially involved in relapse are shown in Figure 3B and Appendix A. The differential analysis revealed 2183 differentially expressed genes (DEG) in RS4;11 in response to Dexa treatment, confirming the sensitivity of the RS4;11 cell line to Dexa. Furthermore, both drugs share, considering both cell lines, 23 potential target genes, although a high amount of exclusively deregulated genes was found for each drug release in the corresponding cell line.

### 3.3. Functional Relapse Characteristics in Response to Each Drug Treatment

Subsequently, an over-representation analysis was performed to investigate the GO terms (biological processes (BPs) and molecular functions (MFs)) associated with relapse due to AraC and Dexa in RS4;11 and SEM (Figure 4). The ORA revealed 66 enriched BPs and 45 enriched MFs in both cell lines. More specifically, in RS4;11, the drug release of Dexa is suggested to involve (transmembrane) protein transport and the mitochondria, whereas the drug release of AraC may influence the cell cycle, translation, chromosome segregation, protein targeting and microtubule organization. The drug release of Dexa in SEM may engage morphogenesis and the microtubule bundle, whereas AraC may encompass antigen presentation and chromosome/chromatin/centromere/histone organization. With respect to molecular functions, the Dexa release suggests a modulation in channel and transporter activities whereas the AraC release may involve phosphate/GTPase activity, ribosome and RNA binding in the RS4;11 cell line. In the SEM cell line, the Dexa release revealed that metal ion/cation/ion binding and transmembrane transporter and symporter may be involved in the relapse process, whereas the AraC release may involve hydrolase/galactosyl activity and the microtubule.

### 3.4. miRNA Enrichment Analysis

We further identified enriched miRNAs in each DEG set from the aforementioned analyses by determining the list of miRNAs whose target genes are significantly enriched within the DEG sets. The biological role of these miRNAs was postulated by linking them to functional and disease annotations via an ORA (Appendix A). Interestingly, the identified miRNAs in both cell lines and drug treatments were significantly associated with primary effusion lymphoma (*p* < 1.39 × 10^−9^), acute myeloid leukemia (*p* < 0.0113) and chronic myeloid leukemia (*p* < 3.29 × 10^−4^). The biological functions of the enriched miRNAs of the AraC-related response were hematopoiesis (*p* < 5.95 × 10^−3^), immune response (*p* < 0.0108), inflammation (*p* < 0.027) and cell proliferation (*p* < 5.45 × 10^−3^). For the Dexa-related response, apoptosis (*p* < 0.0145), inflammation (*p* < 1.27 × 10^−3^) and the tumor suppressor miRNAs (*p* < 6.60 × 10^−3^) were significantly enriched. 

### 3.5. Construction of Relapse-Mediated Gene Regulatory Networks (GRN)

Next, we constructed GRNs that combine transcriptional and post-transcriptional regulatory interactions between the dysregulated genes and the corresponding enriched miRNAs and contextualized them to leukemia (see Methods). Moreover, we ranked the genes/miRNAs according to their node centrality in order to consider their putative mechanistic impact and to characterize the central hub nodes (the “hotspots”) that drive the dynamic regulation of the relapse process after the release of drug pressure, and that therefore potentially act as master regulators. 

The exclusive regulatory modulation in response to AraC treatment revealed, in RS4;11, a GRN based on two miRNAs (hsa-mir-370, hsa-let7e) and three hotspot driver genes (RIMS3, DHX35, HSPA12) (Figure 5A), whereas, in SEM, the GRN consists of three miRNAs (mir-30a, mir-370, mir-34b) and seven hotspot driver genes (LIMCH1, NAV1, CCDC107, SHROOM4, MARCKS, BEX2, PLEKHA6) (Figure 5B). For the GC-sensitive RS4;11 cell line, we constructed a regulatory network comprising 95 hotspot genes and 28 hotspot miRNAs (Figure 5C). In contrast, for the SEM cells, four hotspot miRNAs (mir-31, mir-410, let-7b, let-7c) and six hotspot driver genes (TTC33, DYNC1l1, QSERV, LOXL4, AGPS, MYRIP) were identified (Figure 5D).

### 3.6. Functional Similarity and Integrity between the Relapse–Mediated GRN Genes

We further assessed the biological relevance and the cooperative functional roles of the genes and miRNAs that may be involved in the relapse-mediated GRNs. For each GRN, we computed the functional similarity scores between all gene pairs, including the miRNA target genes. By comparing the resulting score distribution with the similarity score distribution of randomly selected gene pairs, we found that the GRN genes and the miRNA targets have significantly more functional similarity than the randomly selected pairs (*p*-values < 3.9 × 10^−3^, Kolmogorov–Smirnov test), Figure 6. This implies the functional homogeneity and integrity within the GRN genes and miRNAs, representing the molecular interactions and dysregulation mechanisms that may underlie the relapse process associated with each treatment.

## 4. Discussion

In this study, we used the cell lines RS4;11 and SEM isolated from relapsed patients to mimic an unfavorable clinical phenotype (KMT2A (also known as MLL) rearrangement) in adult and childhood pro-B-ALL and to study the effect of the drug release (144 h (+/−)) of the nucleoside analog AraC and the glucocorticoid Dexa. We examined the potential roles of deregulated genes and miRNAs, which may be involved in the tumor relapse for B-ALL, by identifying (1) proliferation and metabolism rates, (2) enriched biological processes (BPs) and molecular function (MF) terms based on DEGs, (3) relapse-mediating regulatory interactions (GRN) and (4) hotspot genes and miRNAs that could potentially drive the relapse mechanisms.

While proliferation increased after a drug pressure release of AraC, it was still reduced after a Dexa release. Metabolic assays revealed minor activity for the AraC treatment and a slight decrease for Dexa. Both measurements clearly showed the corresponding drug sensitivity/resistance of the cell lines. The ORA identified different BPs and MFs terms that could be involved in the B-ALL relapse mechanism after each drug release. For instance, AraC’s release is mainly associated with the BP terms cell cycle, translation and antigen presentation, whereas Dexa’s release is mostly associated with transmembrane-related mechanisms and microtubule bundles. Further, each drug release influences different MFs, such as ribosome and RNA binding for AraC’s release and channel/transporter activities and ion binding for Dexa. 

Unexpectedly, the MF term “mesenchymal stem cell proliferation” is suggested to be triggered by AraC’s release in RS4;11. This specific function was reported before to be associated with the progression of AML [47,48]. In concordance with the fact that Dexa causes cell toxicity [49,50], our enrichment analysis revealed related BPs, such as toxicity and cell mortality [51]. These functions highlight the differences between childhood and adult B-ALL, mainly giving possible insights into the effect of drug release and related tumor progression in adult B-ALL besides Dexa’s potential for severe side effects. 

The molecular differences between the release of both agents become even more evident by constructing their putative relapse-mediated GRNs and identifying the corresponding hotspot genes and miRNAs that likely control the subsequent relapse process. The following is known about the miRNAs highlighted in Figure 5. hsa-mir-370 is linked to the response to chemotherapy [52] and chemosensitivity [53]; hsa-mir-30a is a tumor suppressor [54] found to be mainly associated with better overall survival [55,56,57], poor prognosis [58], early recurrence [59] and response to chemotherapy [60]; hsa-mir-34b plays an essential role regarding tumor progression [61,62] and recurrence [63]; miRNAs hsa-mir-124, hsa-mir-9 and hsa-mir-29a are mainly associated with shorter survival [64], poor prognosis [65,66], GC resistance [67] and a higher amount of blasts in the bone marrow [68]. Furthermore, the hsa-mir-15/16 family is responsible for an insufficient response to chemotherapy and shorter survival [69], a poor prognosis [70], and it is correlated with BCL2 expression, which leads to apoptosis resistance [64]; hsa-mir-410 was previously shown to have a key role in proliferation, colony formation and the apoptosis of ALL cancerous cells [71], and the hsa-let-7 family is known as tumor suppressor miRNAs influencing chemosensitivity [72,73]. 

Interestingly, some of the genes, such as angiomotin (*AMOT*), adenylate kinase 1 (*AK1*), Musashi RNA binding protein (*MSI*), signal transducer and activator of transcription (*STAT2*) and POU class 2 homeobox 2 (*POU2F2*), were reported before as cancer survival markers [74,75,76,77,78,79,80], whereas other genes, such as F-box and WD repeat domain containing 7 (*FBXW7*) and choline kinase alpha (*CHKA*), were found to be regulated by GC treatment [23,71,81]. Myristoylated alanine rich protein kinase C substrate (*MARCKS*) is of high relevance due to being associated with drug resistance [82,83], as well as being a target gene to improve clinical outcome [84,85]. Some genes are known risk markers, such as myosin VIIA And rab interacting protein (*MYRIP*) for infant ALL [86], or alkylglycerone phosphate synthase (*AGPS*) for shorter survival [87].

Study limitations. The main limitation of any study based on in vitro data is the missing confirmation of the insights in an in vivo (animal, xenograft or patient) setting. At this moment, some confidence can be based on the plausibility of many of our findings, given the literature-based discussion above. Within the in vitro setting, it would of course be useful to test far more cell lines, drugs and timings of drug application; however, resources are limited. Regarding the bioinformatics analyses, an array of straightforward canonical analyses was performed, and insightful network-biology approaches were executed. However, the choice of further potential analyses is vast. Then again, the basic insights of our analyses matched a lot of knowledge already reported in the literature, so alternative analyses are expected to yield results that, to a sufficient degree, are expected to match the ones reported. 

## 5. Conclusions

In conclusion, we revealed molecular differences in drug release and possible subsequent relapse-mediated mechanisms in response to two cytostatic agents (AraC and Dexa) in two B-ALL cell line models, highlighting candidate genetic drivers of the ALL progression with respect to the age of onset (childhood and adults). The constructed relapse-mediated GRNs for each agent were assessed for their functional homogeneity and literature evidence. This regulatory analysis helped us to spot genes and miRNAs that are exclusively deregulated by the two agents in the selected settings. Further wet lab experiments are warranted to test the usefulness of these genes and miNRAs as therapeutic targets and/or prognosis indicators in B-ALL. They may ultimately be used to develop a superior application protocol to best treat the patient.

## Figures and Tables

**Figure 1 genes-13-01240-f001:**
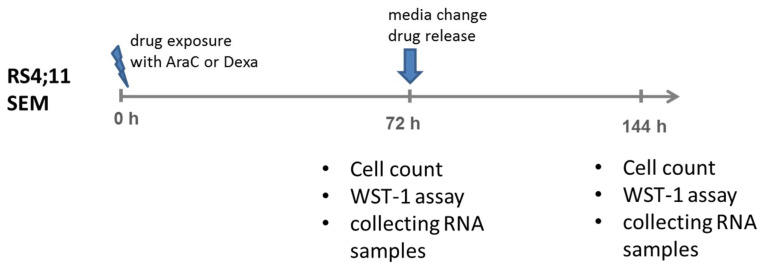
Schematic diagram for the experimental setup to study relapse mechanisms and identify biomarkers for relapse in pro-B-ALL. Two settings were chosen: (1) drug exposure for 72 h, followed by (2) no drug exposure (“drug release”) for an additional 72 h, noted here as 144 h (+/−). In the experimental readout, trypan-blue staining (cell count), WST-1 assay (cell count and metabolism analysis) and sample collection for following RNA-Seq of RS4;11 and SEM were performed.

**Figure 2 genes-13-01240-f002:**
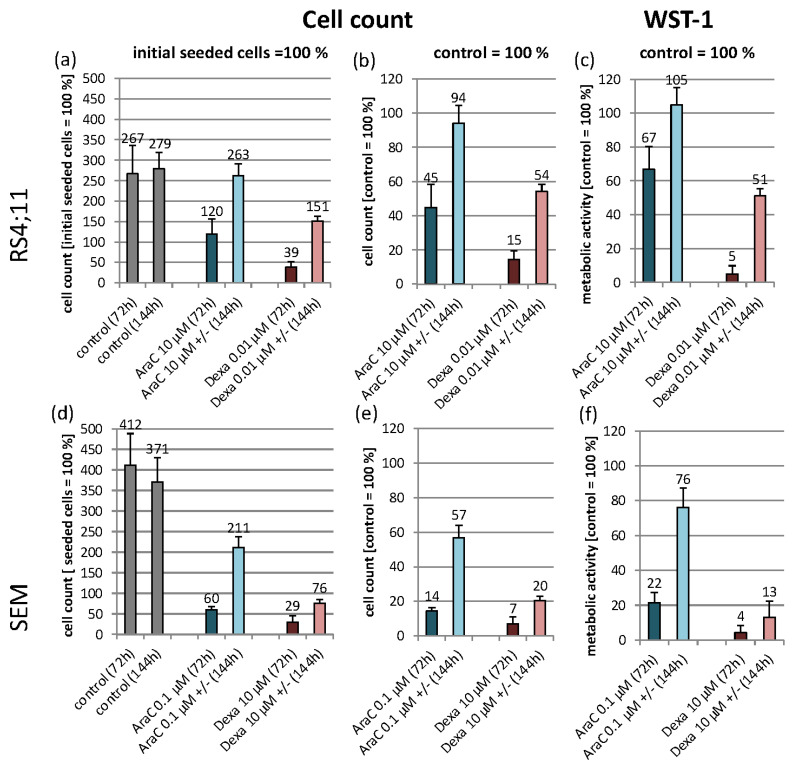
Cell count and metabolic activity in RS4;11 (**a**–**c**) and SEM (**d**–**f**) in response to AraC and Dexa for 72 h. After 72 h, drugs were released and cells cultivated for another 72 h (144 h+/−). Cell count (**a**,**b**,**d**,**e**) and WST-1 proliferation assay (**c**,**f**) were performed after 72 h and 144 h. The control cells and drug exposure times were compared with the Students’ *t*-test at significance levels of * *p* ≤ 0.05, ** *p* ≤ 0.01, *** *p* ≤ 0.001, [*n* ≥ 3].

**Figure 3 genes-13-01240-f003:**
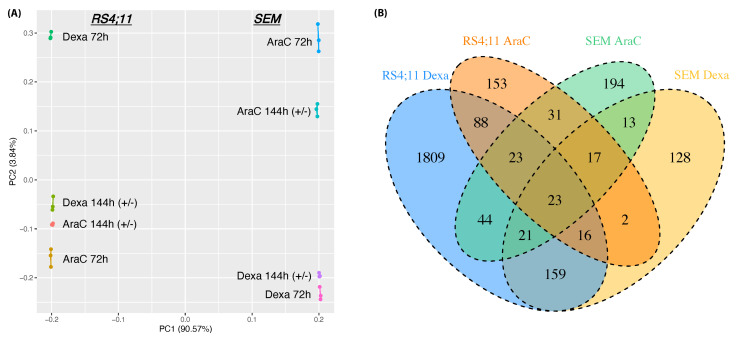
Dissimilarity of the transcriptional response of Dexa and AraC when applied to SEM and RS4;11. (**A**) PCA of whole transcriptome profiles of AraC and Dexa applications in the cell lines. (**B**) Venn diagram showing the differentially expressed genes (DEGs) of each drug treatment in each cell line. The experiments were carried out in biological triplicates.

**Figure 4 genes-13-01240-f004:**
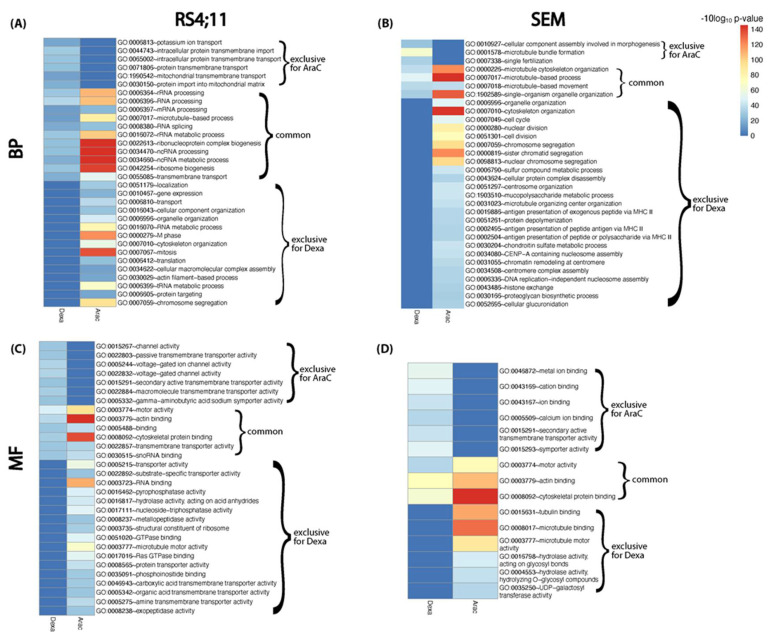
Over-represented biological processes and molecular functions for the cell lines RS4;11 (**A**) and for SEM (**B**) shown separately for the exclusive patterns for the drug releases of AraC and Dexa, as well as the common ones. The molecular functions for RS4;11 (**C**) and SEM (**D**) in response to the drug release of AraC and Dexa differentiating between common and exclusive over-representation.

**Figure 5 genes-13-01240-f005:**
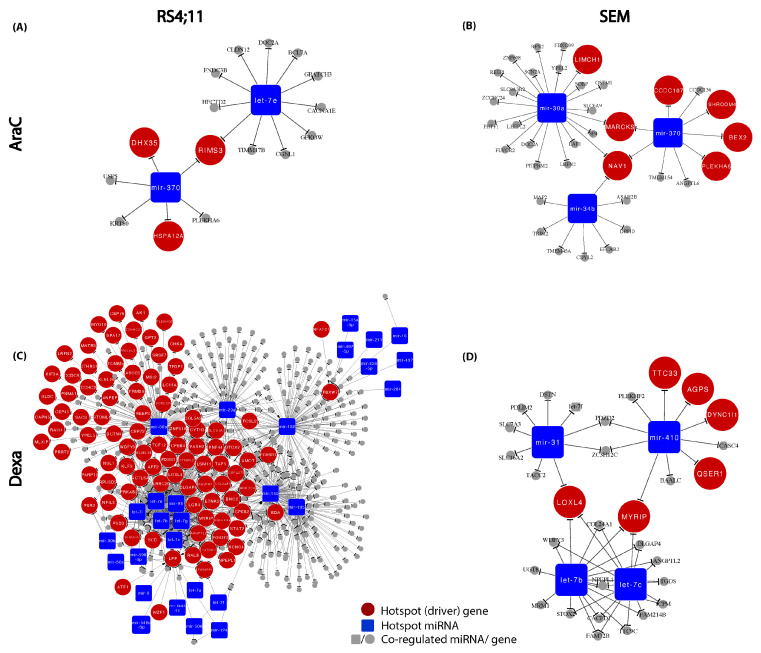
GRNs in response to AraC and Dexa in RS4;11 and SEM. Large red nodes represent the hotspot (driver) genes, blue squares indicate hotspot miRNAs and grey nodes denote the co-regulated miRNAs/genes for (**A**) AraC in RS4;11, (**B**) AraC in SEM, (**C**) Dexa in RS4;11 and (**D**) Dexa in SEM.

**Figure 6 genes-13-01240-f006:**
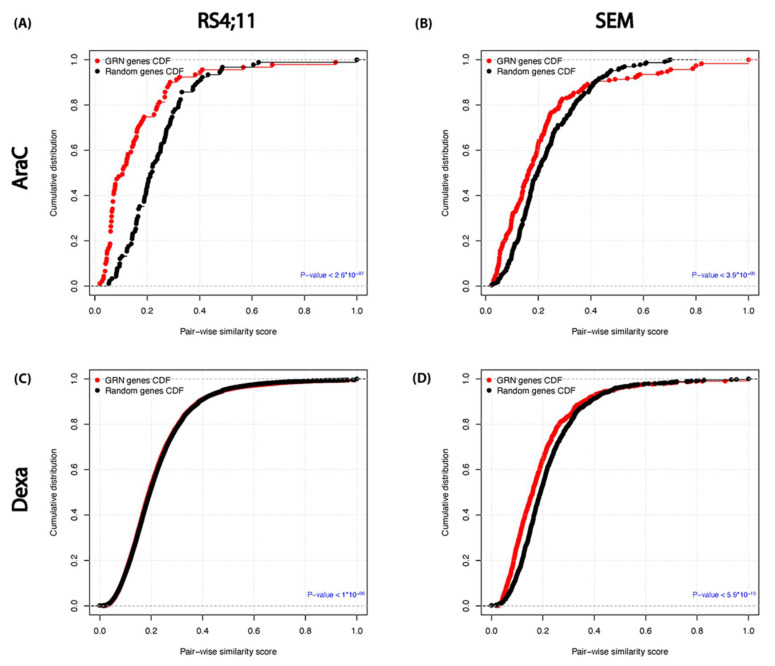
The functional homogeneity within each drug–GRN in the two examined cell lines. For each GRN, the cumulative distribution of GO functional semantic scores of gene pairs of the GRN (red) versus randomly selected genes (black) are depicted for (**A**) AraC in RS4;11, (**B**) AraC in SEM, (**C**) Dexa in RS4;11 and (**D**) Dexa in SEM. The corresponding *p*-values were calculated using the Kolgomorov–Smirnov test. The GRNs were constructed with the TFmiR web service (41) for each drug treatment based on the DEG set and the corresponding enriched miRNA set, and visualized using Cytoscape V3.3.0 [45].

## Data Availability

The RNA-Seq data are provided in Supplementary File S2.

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
