# Peer review of "Molecular Characterization of the Response to Conventional Chemotherapeutics in Pro-B-ALL Cell Lines in Terms of Tumor Relapse"

_genes, 2022, doi:10.3390/genes13071240_

Round 1
Reviewer 1 Report
In this article, Yvonne Saara Gladbach et al analyzed molecular profiles of pro-B-ALL cell lines after chemotherapy treatments. The manuscript covers several aspects of the subject and is well organized.
However, the paper is too long and some paragraphs need to be rewritten and reduction (more than 50%) such introduction and materials and methods.
For the clarity of the paper, the authors can be summarized all the data in a table.
Author Response
Reviewer 1
1.Introduction: By the end of the introduction, the aim of the study should be clear, and the readers should be interested in continuing reading the manuscript to find the results. In the sense, the results of the current work should not be mentioned in the section (ll 109-116). The appropriate place for this paragraph is in the conclusion section
We thank our reviewer for this comment. We followed his suggestion and removed the section completely since its main contents are already included in the conclusion section.
- Section 2.2;line 131: What was the dilution factor applied in this context.
For all 10µM concentration we used a 1:10 dilution, for 0,1µM 1:100 and for 0,01µM 1:1000.
This has been added also to the materials and methods
- Section 2.6: line 189: please add “gene” to the long name of DEG
Done
- line 192: Supplement table 7 was not included in the suppl. File. Also, its place is not chronologically appropriate.
Thank you for this comment. This was actually a typo. We have only 6 supplement tables in supplementary file 1. The read counts/ processed datasets are available in supplementary file 2. We have changed the text accordingly.
- line 249: “the control cells after 72h respectively 144h as 100% in Figure 2….” The sentence was not clear.
We changed the sentence to the following:
In order to quantify the biological effects of the two drug applications on proliferation rate and metabolism, we set (1) the initially seeded cells as 100% in Figure 2(A,D) and (2) the control cells as 100% (readout 72h and 144h) in Figure 2(B,C,E,F). In general, incubation with AraC and Dexa causes, in both cell lines, a reduced proliferation rate compared to the control cells after 72h.
- It was hard to follow the authors´elaboration and comments on figure 4 and 5 due to the low resolution of the figures.
We provide all figures in higher resolutions and this point should be tackled after processing the manuscript by the journal submission system.
- line 345:”Marcks” should be written in its long name at its first mention.
Done. We changed the sentence to the following:
Myristoylated Alanine Rich Protein Kinase C Substrate (MARCKS) is of high relevance due to being associated with drug resistance [49,50] as well as being a target gene to improve clinical outcome [51,52].
- Section 3.51 and 3.5.2: the authors elaborated on the findings and discussed them as in case of discussion with citations. This section should be revised, and the authors should keep only the presentation of their findings without citations. The comments and explanation of the findings should be moved to the discussion section.
We thank our reviewer for this valuable comment. We followed his idea and moved the parts of 3.5.1 and 3.5.2 to the discussion section.
- lines 354-356: all the gene acronyms should be italicized to match the standards of HUGO for gene nomenclatures. Also, their long names should be mentioned in the text.
Done. We italicized all gene names and added the long name for each of the genes.
- After enrichment of the discussion with the comments and citations mentioned above, the authors should provide the study limitation(s) by the end of the discussion.
Done. We now provide a separate section at the end on the limitations and caveats of our study.
Minor comments: the authors should avoid duplicating the words in case the acronyms contain these words. (For example: GRN networks”, “PCA analysis”
-We followed the suggestion of our reviewer wherever possible.
Reviewer 2 Report
MLL has been renamed in WHO classification 2017 revision as [K]-specific MethylTransferase 2A or KMT2A.
Author Response
Reviewer 2
MLL has been renamed in WHO classification 2017 revision as [K]-specific MethylTransferase 2A or KMT2A.
We thank our reviewer for this notice. Now we changed MLL to KMT2A in our text.
Reviewer 3 Report
Overall, the idea of the work is interesting. The work is well done with the quality control measures applied. The findings of the work could have merit in the related field. Just a few concerns should be addressed.
1- Introduction
By the end of the introduction, the aim of the study should be clear, and the readers should be interested in continue reading the manuscript to find the results. In this sense, the results of the current work should not be mentioned in this section (lines 109-116). The appropriate place for this paragraph is in the conclusion section.
2- Section 2.2; line131: What was the dilution factor applied in this context.
3- Section 2.6; line 189: please add “gene” to the long name of DEG
4- Line 192: Supplemental Table 7 was not included in the Suppl. file. Also, its place is not chronologically appropriate.
5- Line 249: “the control cells after 72h respectively 144h as 100% in Figure 2…” this sentence was not clear.
6- It was hard to follow the authors' elaboration and comments on Figures 4 and 5 due to the low resolution of the figures.
7- Line 345: “MARCKS” should be written in its long name at its first mention.
8- Sections 3.51 and 3.5.2: The authors elaborated on the findings and discussed them as in case of discussion with citations. These sections should be revised, and the authors should keep only the presentation of their findings without citations. The comments and explanation of the findings should be moved to the discussion section.
9- Lines 354-356: All the gene acronyms should be italicized to match the standards of HUGO for gene nomenclatures. Also, their long names should be mentioned in the text.
10 – After enrichment of the discussion with the comments and citations mentioned above, the authors should provide the study limitation(s) by the end of the discussion.
Minor comments
The authors should avoid duplicating the words in case the acronyms contain these words. For example:
- Line 220: The “GRN networks”.
- Lines 274 and 293: “PCA analysis”.
Author Response
Reviewer 1
1.Introduction: By the end of the introduction, the aim of the study should be clear, and the readers should be interested in continuing reading the manuscript to find the results. In the sense, the results of the current work should not be mentioned in the section (ll 109-116). The appropriate place for this paragraph is in the conclusion section
We thank our reviewer for this comment. We followed his suggestion and removed the section completely since its main contents are already included in the conclusion section.
- Section 2.2;line 131: What was the dilution factor applied in this context.
For all 10µM concentration we used a 1:10 dilution, for 0,1µM 1:100 and for 0,01µM 1:1000.
This has been added also to the materials and methods
- Section 2.6: line 189: please add “gene” to the long name of DEG
Done
- line 192: Supplement table 7 was not included in the suppl. File. Also, its place is not chronologically appropriate.
Thank you for this comment. This was actually a typo. We have only 6 supplement tables in supplementary file 1. The read counts/ processed datasets are available in supplementary file 2. We have changed the text accordingly.
- line 249: “the control cells after 72h respectively 144h as 100% in Figure 2….” The sentence was not clear.
We changed the sentence to the following:
In order to quantify the biological effects of the two drug applications on proliferation rate and metabolism, we set (1) the initially seeded cells as 100% in Figure 2(A,D) and (2) the control cells as 100% (readout 72h and 144h) in Figure 2(B,C,E,F). In general, incubation with AraC and Dexa causes, in both cell lines, a reduced proliferation rate compared to the control cells after 72h.
- It was hard to follow the authors´elaboration and comments on figure 4 and 5 due to the low resolution of the figures.
We provide all figures in higher resolutions and this point should be tackled after processing the manuscript by the journal submission system.
- line 345:”Marcks” should be written in its long name at its first mention.
Done. We changed the sentence to the following:
Myristoylated Alanine Rich Protein Kinase C Substrate (MARCKS) is of high relevance due to being associated with drug resistance [49,50] as well as being a target gene to improve clinical outcome [51,52].
- Section 3.51 and 3.5.2: the authors elaborated on the findings and discussed them as in case of discussion with citations. This section should be revised, and the authors should keep only the presentation of their findings without citations. The comments and explanation of the findings should be moved to the discussion section.
We thank our reviewer for this valuable comment. We followed his idea and moved the parts of 3.5.1 and 3.5.2 to the discussion section.
- lines 354-356: all the gene acronyms should be italicized to match the standards of HUGO for gene nomenclatures. Also, their long names should be mentioned in the text.
Done. We italicized all gene names and added the long name for each of the genes.
- After enrichment of the discussion with the comments and citations mentioned above, the authors should provide the study limitation(s) by the end of the discussion.
Done. We now provide a separate section at the end on the limitations and caveats of our study.
Minor comments: the authors should avoid duplicating the words in case the acronyms contain these words. (For example: GRN networks”, “PCA analysis”
We followed the suggestion of our reviewer wherever possible.